# Diet and Risk of Non-Alcoholic Fatty Liver Disease, Cirrhosis, and Liver Cancer: A Large Prospective Cohort Study in UK Biobank

**DOI:** 10.3390/nu14245335

**Published:** 2022-12-15

**Authors:** Wen Guo, Xinyuan Ge, Jing Lu, Xin Xu, Jiaxin Gao, Quanrongzi Wang, Ci Song, Qun Zhang, Chengxiao Yu

**Affiliations:** 1Health Management Center, The First Affiliated Hospital of Nanjing Medical University, Nanjing 210029, China; 2Department of Epidemiology, China International Cooperation Center on Environment and Human Health, Center for Global Health, School of Public Health, Nanjing Medical University, Nanjing 211166, China; 3Department of Radiology, The First Affiliated Hospital of Nanjing Medical University, Nanjing 210029, China; 4Research Units of Cohort Study on Cardiovascular Diseases and Cancers, Chinese Academy of Medical Sciences, Beijing 100000, China; 5The Affiliated Suzhou Hospital of Nanjing Medical University, Suzhou Municipal Hospital, Gusu School, Nanjing Medical University, Suzhou 215008, China

**Keywords:** dietary pattern, non-alcoholic fatty liver disease, cirrhosis, liver cancer, cohort study

## Abstract

Background and Aims: Epidemiological evidence has shown the association between nutritional habits and liver disease. However, results remain conflicting. This study investigated the influence of dietary factors on the risk of incident non-alcoholic fatty liver disease (NAFLD), cirrhosis, and liver cancer. Methods: Data from the UK Biobank database were analyzed (*n* = 372,492). According to baseline data from the food frequency questionnaire, two main dietary patterns (Western and prudent) were identified using principal component analysis. We used cox proportional hazards models to explore the associations of individual food groups and dietary patterns with NAFLD, cirrhosis, and liver cancer. Results: During a median follow-up of 12 years, 3527 hospitalized NAFLD, 1643 cirrhosis, and 669 liver cancer cases were recorded among 372,492 participants without prior history of cancer or chronic liver diseases at baseline. In multivariable adjusted analysis, participants in the high tertile of Western dietary pattern score had an 18% (95%CI = 1.09–1.29), 21% (95%CI = 1.07–1.37), and 24% (95%CI = 1.02–1.50) higher risk of incident NAFLD, liver cirrhosis, and liver cancer, respectively, compared with the low tertile. Participants in the high tertile of prudent scores had a 15% (95%CI = 0.75–0.96) lower risk of cirrhosis, as compared with those in the low tertile. In addition, the higher consumption of red meat and the lower consumption of fruit, cereal, tea, and dietary fiber were significantly associated with a higher risk of NAFLD, cirrhosis, and liver cancer (*p*_trend_ < 0.05). Conclusions: This large prospective cohort study showed that an increased intake of food from the Western dietary pattern could be correlated with an increased risk of chronic liver diseases, while the prudent pattern was only correlated with a reduced liver cirrhosis risk. These data may provide new insights into lifestyle interventions for the prevention of chronical liver diseases.

## 1. Introduction

In recent decades, chronic liver diseases have relentlessly risen to be an important cause of morbidity and mortality from noncommunicable diseases in both developed and developing countries [1,2]. Meanwhile, rapid changes in lifestyle and aging population are generally believed to account for huge changes in the pattern of risk factors for chronic liver disease, and health priorities are no longer limited to viral liver diseases. Consequently, identifying modifiable risk factors is critical to prevent the development of chronic liver diseases and for early intervention.

The new epidemic in chronic liver disease is related to the burden of non-alcoholic fatty liver disease (NAFLD), paralleling the worldwide increase in obesity. The prevalence of NAFLD is estimated at about 24% globally, with considerable variability from different countries [3]. NAFLD individuals have a higher risk of developing cirrhosis and liver cancer [4]. Nowadays, it is generally accepted that diet plays a pivotal role in the pathogenesis of chronic liver diseases, but it is also the cornerstone approach for its management [5,6]. Recent studies have indicated that a diet rich in sugar, saturated fatty acids, and cholesterol promotes the pathogenesis and development of NAFLD [7,8]. However, a diet rich in fruit, protein, polyunsaturated fatty acids, and vegetables is associated with decreased risk of NAFLD [9]. In patients with type 2 diabetes mellitus (T2DM), liver fibrosis was negatively associated with the adherence to a Mediterranean diet [10]. In a study conducted in U.S. adults, there was no relationship between the alternate Mediterranean diet and the risk of hepatocellular carcinoma (HCC) development [11]. In contrast, a systematic review showed that the Urban Prudent Dietary Pattern was significantly correlated with a reduced risk of HCC [12]. Regular diets often consist of complex mixtures of nutrients and foods which may influence each other, suggesting that entire diets as dietary patterns may give us a more comprehensive understanding of the link between chronic diseases and diet [13]. However, in the literature surrounding dietary patterns associated with different types of liver diseases from recent years, the findings were inconsistent due to limited study designs and sample sizes. It is necessary to study the multiple dietary factors with chronic liver diseases based on a large prospective population.

Thus, in order to provide a more comprehensive insight into the overall effect of dietary patterns on the risk of chronic liver diseases, we conducted a longitudinal analysis to evaluate the impact of dietary patterns and individual food groups on the risk of chronic liver diseases including NAFLD, liver cirrhosis, and liver cancer.

## 2. Materials and Methods

### 2.1. Study Design and Population

UK Biobank was designed to provide a resource for the investigation of genetic, environmental, and lifestyle factors associated with the risk of human diseases. The UK Biobank is a large-scale prospective cohort study, which enrolled over 500,000 individuals aged 40–70 from 2006 to 2010 [14]. All participants who provided written informed consent were interviewed using a touch-screen questionnaire to collect social demographic, lifestyle, and health-related information, as well as completed a range of anthropometric and physiological measurements. The UK Biobank obtained ethical approval from the NHS National Research Ethics Service.

In our study, participants with incomplete dietary information (*n* = 43,528) and other covariables (*n* = 85,069) were excluded. Moreover, 1326 study subjects with prevalent NAFLD, cirrhosis, liver cancer, and other basic liver diseases at baseline (such as viral hepatitis, alcoholic liver diseases, liver failure, etc.) were excluded (Appendix A). After the above exclusions, there were 372,492 participants in our final analyses (Appendix A).

### 2.2. Assessment of Dietary Intake

Information on intake of fresh fruit, dried fruit, raw vegetables, cooked vegetables, processed meat, oily fish, non-oily fish, lamb, pork, poultry, beef, cheese, bread, cereal, and tea was requested in a Food Frequency Questionnaire (FFQ) (category 100052) from the UK Biobank. Appendix A shows the coding and classification of each food group. We calculated fiber scores according to information on fruit, vegetables, types and intake of bread, and types and intake of cereal, which were derived from the touch screen questionnaire, as described in previous studies [15].

### 2.3. Assessment of Dietary Patterns

Principal component analysis (PCA) was conducted using varimax rotation on the 15 food items to identify dietary patterns, which was standardized in advance [16]. The quantity of retaining factors were determined by the scree plot, eigenvalues (>1), and interpretability [17]. A higher rotated factor loading value of food group represented a stronger association with dietary pattern, and when the absolute value was ≥ |0.30|, we considered that the corresponding food was significantly correlated with the dietary pattern [18]. Finally, we identified dietary habits as the following patterns: prudent pattern and Western pattern [19], and the continuous dietary pattern scores were categorized in tertiles.

### 2.4. Ascertainment of Outcomes

In the UK Biobank, we obtained follow-up data on liver-related events and mortality through electronic connection to the in-hospital admissions and cancer registry in England, Wales, and Scotland. Participants were followed-up from the date of attendance at the assessment center until the date of death, date of diagnosis, or last date of follow-up (30th September 2021 for England, 28th February 2018 for Wales, and 31st July 2021 for Scotland), whichever came first. The outcomes of interest were incident NAFLD, cirrhosis, and liver cancer, which were coded using the 10th Revision of the International Classification of Diseases (ICD-10). Incident NAFLD and liver cancer were defined as ICD-10 K76.0 and C22, respectively. Incident cirrhosis was defined by the combination of a series of diseases [20]: K70.2 (alcoholic fibrosis and sclerosis of the liver), K70.3 (alcoholic cirrhosis), K70.4 (alcoholic hepatic failure), K74.0 (hepatic fibrosis), K74.1 (hepatic sclerosis), K74.2 (hepatic fibrosis with hepatic sclerosis), K74.6 (other and unspecific cirrhosis of liver), K76.6 (portal hypertension), or I85 (esophageal varices) (Appendix A). Controls were free of above liver diseases at baseline.

### 2.5. Covariates

We included several socio-demographic and behavioral risk factors as covariates, which could confuse the effect of dietary intake on liver-related events potentially. Age was calculated using date of attending assessment center and birth date. Race was grouped as “white ethnicity” and “other”. Townsend deprivation index was divided into four groups by quartile. Education level was defined as “college or university degree” or “other levels”. Smoking and drinking status were classified into never, former, and current. Regular exercise indicates whether a person met the guidelines of 75 min of vigorous activity per week or 150 min of moderate activity. Body mass index (BMI) was calculated by weight and height (kilogram/square meter) and classified as normal (<25), overweight (25–29.9), and obesity (≥30) for each participant. Diabetes was defined on the basis of self-reported diagnosis of diabetes or use of diabetes medication.

### 2.6. Statistical Analysis

We used cox proportional hazards model to assess associations of liver-related event risk with dietary patterns. The hazard ratios (HRs) and corresponding 95% confidence intervals (CIs) were estimated with adjustment for age, sex, race, education level, Townsend deprivation index, drinking status, smoking status, exercise, BMI, and diabetes. *p* values for trend were assessed by fitting categories as continuous variables in models. Schoenfeld residuals were used to test the proportional hazards assumption.

In order to examine whether the associations between the dietary patterns and liver-related events varied by subgroups, we conducted stratified analyses by age (years, <60, ≥60), sex (male, female), ethnic background (white, non-white), Townsend deprivation index (below, above median), education (college or university degree or not), regular exercise (yes, no), smoking status (never, former, current), drinking status (never, former, current), BMI (kg/m^2^, <25, 25–29.9, ≥30), and diabetes (yes, no). We utilized χ^2^-based Cochrane’s Q test to evaluate the heterogeneity between above stratified results.

Several sensitivity analyses were conducted to evaluate the stability of the results: (1) restricting analyses in subjects who reported that they had not made any major changes to their diet in the past 5 years (*n*  =  229,374); (2) excluding the subjects with 3 years or less of follow-up; (3) excluding excessive alcohol consumption ( ≥ 30 g/day and ≥20 g/day for men and women, respectively, *n* = 76,203).

All analyses were conducted using R 4.1.0 (R Foundation for Statistical Computing, Vienna, Austria). *p*-values were two-sided, and *p* < 0.05 was considered statistically significant.

## 3. Results

During a median follow-up of 12 years, 3527 hospitalized NAFLD, 1643 cirrhosis, and 669 liver cancer cases were recorded among 372,492 participants without prior history of chronic liver diseases at baseline. Table 1 shows the baseline characteristics of the participants according to liver diseases. The participants who developed NAFLD were of low income level, less educated, physically inactive, more likely to be current smokers, metabolically unhealthy, and fatter. The similar results were also observed in participants who developed cirrhosis and liver cancer. Additionally, participants who developed cirrhosis and liver cancer tended to be male.

We identified two dietary patterns by principal component analysis, which were labeled as the prudent pattern and Western pattern, explaining a cumulative variance of 25% in the consumption of 15 food groups (Table 2). The prudent pattern indicated a high consumption of salad raw vegetables, cooked vegetables, fresh fruit, dried fruit, oily fish, and non-oily fish. The Western pattern was characterized by a high consumption of processed meat, poultry, beef, lamb mutton, and pork. Correlations between different dietary components are shown in Appendix A.

The risks of chronic liver diseases were analyzed according to tertiles of baseline pattern scores (Figure 1). The risks of chronic liver diseases were significantly elevated in participants in the high tertile of the Western pattern score compared to those in the low tertile, with adjusted HRs of 1.18 (95%CI 1.09–1.29) for NAFLD, 1.21 (95%CI 1.07–1.37) for cirrhosis, and 1.24 (95%CI 1.02–1.50) for liver cancer in the fully adjusted model. Compared with the tertile 1 of the prudent pattern score, tertile 3 was significantly associated with a lower prevalence of cirrhosis (HR = 0.85, 95%CI 0.75–0.96) after full adjustment. However, no significant associations of prudent pattern with NAFLD and liver cancer were observed. In addition, the association trend between specific types of cirrhosis and dietary patterns was consistent (Appendix A).

Hazard ratios for the risks of different liver diseases by individual food groups were listed in Appendix A. A higher intake of red meat (HR = 1.18, 95%CI 1.03–1.35) was independently significantly correlated with a higher risk of NAFLD. However, a higher intake of fruit (HR = 0.89, 95%CI 0.81–0.98), cereal (HR = 0.81, 95%CI 0.74–0.89), and tea (HR = 0.85, 95%CI 0.77–0.94) were significantly correlated with a lower risk of NAFLD. Similar results were also observed in participants who developed cirrhosis and liver cancer. Additionally, a lower intake of cheese (HR = 0.88, 95%CI 0.77–0.99) and a higher intake of poultry (HR = 1.33, 95%CI 1.10–1.61) significantly increased the risk of NAFLD.

Each 5-g/d increment of total dietary fiber was correlated with a 7% (HR = 0.93, 95%CI 0.91–0.96), 14% (HR = 0.86, 95%CI 0.91–0.96), and 9% (HR = 0.91, 95%CI 0.85–0.98) decreased risk of NAFLD, cirrhosis, and liver cancer, respectively (Appendix A). For specific fiber sources, fiber from bread and cereal (HR _per 5 g/d_ = 0.81, 95%CI 0.76–0.87), fruits (HR _per 5 g/d_ = 0.94, 95%CI 0.89–0.99), but not vegetables (HR _per 5 g/d_ = 1.01, 95%CI 0.96–1.06) were inversely correlated with NAFLD. Moreover, fiber from bread and cereal was inversely associated with cirrhosis (HR _per 5 g/d_ = 0.70, 95%CI 0.64–0.77) and liver cancer (HR _per 5 g/d_ = 0.78, 95%CI 0.68–0.90). However, a null significant association was found between fiber from fruits and vegetables, cirrhosis, and liver cancer.

Associations of dietary patterns by stratification in the UK Biobank are shown in Appendix A, and the effects on the risk of liver disease were generally similar in different subgroups. Sensitivity analyses were performed by excluding participants who reported changing their diet, with less than 3 years of follow-up and with excessive alcohol consumption (Appendix A) and produced consistent results.

## 4. Discussion

In this large prospective cohort study, a baseline high consumption of breakfast cereals, tea, fruits, and dietary fiber, as well as a low intake of red meat and processed meat, were significantly associated with the future risk of chronic liver diseases including NAFLD, cirrhosis, and liver cancer. For specific fiber sources, there were negative associations of fiber from bread and breakfast cereals with NAFLD, cirrhosis, and liver cancer. Dietary pattern analysis offers an approach to examine the relationship between whole-diet and diseases, which can investigate the interactions or combined effects of various dietary components on disease risk. Our prospective study further found that participants with the prudent dietary pattern had a reduced risk of cirrhosis, while participants with the Western pattern had an increased risk of NAFLD, cirrhosis, and liver cancer. The above findings reveal potential implications for chronic liver diseases prevention and management.

NAFLD is a major cause of chronic liver disease [21]. The progression of NAFLD can range from benign presentation of simple steatosis to the more severe form of the disease, steatohepatitis (NASH), which can advance to end-stage liver disease, resulting in significant health care resource use and decreased health-related quality of life. Dietary factors have been recognized as a predominant cause of the occurrence and progression of NAFLD. Similar to a cross-sectional study and a case–control study [22,23], our prospective cohort study showed that a higher consumption of red meat and processed meat significantly increased the future risk of NAFLD. Lee JH et al. demonstrated that there were negative associations of intake of milk and other dairy products with incident NAFLD [24]. In our study, we also found the inverse association between cheeses and incident NAFLD. Fruit and breakfast cereals, which are basic components of our daily diet, have been shown to reduce the risk of NAFLD [25]. This is consistent with our study, which found that fruit and breakfast cereals were inversely associated with incidence of NAFLD. However, we did not find the correlation between higher intake of vegetables and NAFLD risk, which is inconsistent with previous studies [26]. Not all vegetables are correlated with the reduction in chronic diseases risk, due to different components and bioactive phytochemicals in various vegetables [27]. In fact, vegetables which contain higher levels of dietary fiber may reduce the risk of NAFLD [26]. In the present study, we further explored the relationship between fiber sources and risk of NAFLD. We further found that fiber from bread, cereals, and fruits, but not vegetables, was inversely correlated with the risk of NAFLD, consistent with previous studies about diet and risk of incident lung cancer [28]. Given that regular diets are made up of complex combinations of foods and nutrients that may act independently or interact, we investigated the association between dietary patterns and risk of NAFLD. Our findings regarding dietary patterns are consistent with previous studies suggesting that Western dietary patterns are positively correlated with risk of NAFLD [29,30].

Liver cirrhosis is common worldwide and may result from different causes, such as obesity, chronic HBV or HCV infection, alcohol drinking, and NAFLD [31]. The complications of liver cirrhosis often lead to heavy economic burdens, impaired quality of life, and high mortality. Liver cirrhosis is also the major cause of liver cancer. Recent studies suggested that Western diet can accelerate the progression of chronic liver disease and subsequent hepatic fibrosis in rodent models [32]. The International Society for Hepatic Encephalopathy and Nitrogen Metabolism Consensus recommended that diets rich in vegetables, dairy protein, and fiber may be beneficial for liver cirrhosis patients [33]. However, evidence regarding the risk of liver cirrhosis with dietary factors and dietary patterns in humans has never been reported. The present study indicated that a high consumption of red meat and processed meat increased the risk of liver cirrhosis, while a high intake of fruits, cereal, bread, tea, and dietary fiber showed the opposite effect. The novelty of our findings was that the prudent dietary pattern reduced the risk of liver cirrhosis, whereas the Western pattern increased the risk of liver cirrhosis. Indeed, our results provide evidence for the management of people at high risk of cirrhosis.

Liver cancer is of high prevalence and poor long-term clinical outcome, bringing serious public health problems. Besides chronic infections of HBV and HCV, epidemiologic and clinical evidence have shown an association of diet with the risk of incident liver cancer. Evidence showed that a higher intake of red meat and processed meat may contribute to a higher risk of liver cancer [8]. In contrast, high consumption of fruits, vegetables, cereal, and tea appeared to have a favorable effect on liver cancer [21]. However, other studies indicated that the evidence on the association of liver cancer with red meat, processed meat, fruits, and green tea were ‘insufficient’ [34,35,36]. In our study, we observed a positive association of red meat and an inverse association of fruit, cereal, and tea with incident liver cancer. In addition, an increased intake of total fiber, especially cereal fiber, was possibly associated with a reduced risk of liver cancer, while there seemed to be no significant association of fiber from fruit with liver cancer risk [37], consistent with our study. Assessing the relationship between dietary patterns and health outcomes may provide more information about the effect of dietary patterns on health. Our present study showed that the Western pattern was somewhat more strongly correlated with risk of incident liver cancer, while there was a null association of the prudent dietary pattern with incident liver cancer. However, we further found that adherence to fruit, cereal, and tea may reduce liver cancer risk. More studies are needed to confirm this finding.

The precise mechanisms underpinning the association of diet factors and dietary patterns with incident NAFLD, cirrhosis, and liver cancer remain unclear, but could involve oxidative stress, chronic inflammation, altered gut microbiota, insulin sensitivity, and lipid peroxidation [38]. Further studies are needed on the causative role of diet factors and dietary patterns in the physiopathologic mechanism of chronic liver disease.

The strengths of our study include its prospective design, a large sample size, standardized high-quality data collection, and extensive adjustment for potential confounders. Sensitivity analysis was carefully executed to minimize the confounding bias by excluding participants who changed their diet to alleviate this restriction, or those with 3 years or less of follow-up; the results remained virtually unchanged. Anyhow, our study has several limitations. First, NAFLD, liver cirrhosis, and liver cancer were diagnosed using the ICD-10. Liver biopsy is considered the gold standard for the detection of liver steatosis or liver fibrosis but is invasive and costly. Second, participants in our study were from the UK Biobank, which enrolled participants aged 40–69; thus, the findings may not be generalizable to other age or race/ethnicity groups. Third, although we made extensive adjustments for the potential confounders, some residual or unmeasured confounding parameters cannot be completely excluded.

In conclusion, the results of our study, using data from a large perspective cohort study, demonstrated that participants with a Western dietary pattern were more likely to have a higher risk of NAFLD, cirrhosis, and liver cancer, whereas participants with a prudent dietary pattern were more likely to have a lower risk of cirrhosis. Thus, this study suggests that the adherence to a recommended diet pattern will be beneficial for individuals with chronic liver diseases.

## Figures and Tables

**Figure 1 nutrients-14-05335-f001:**
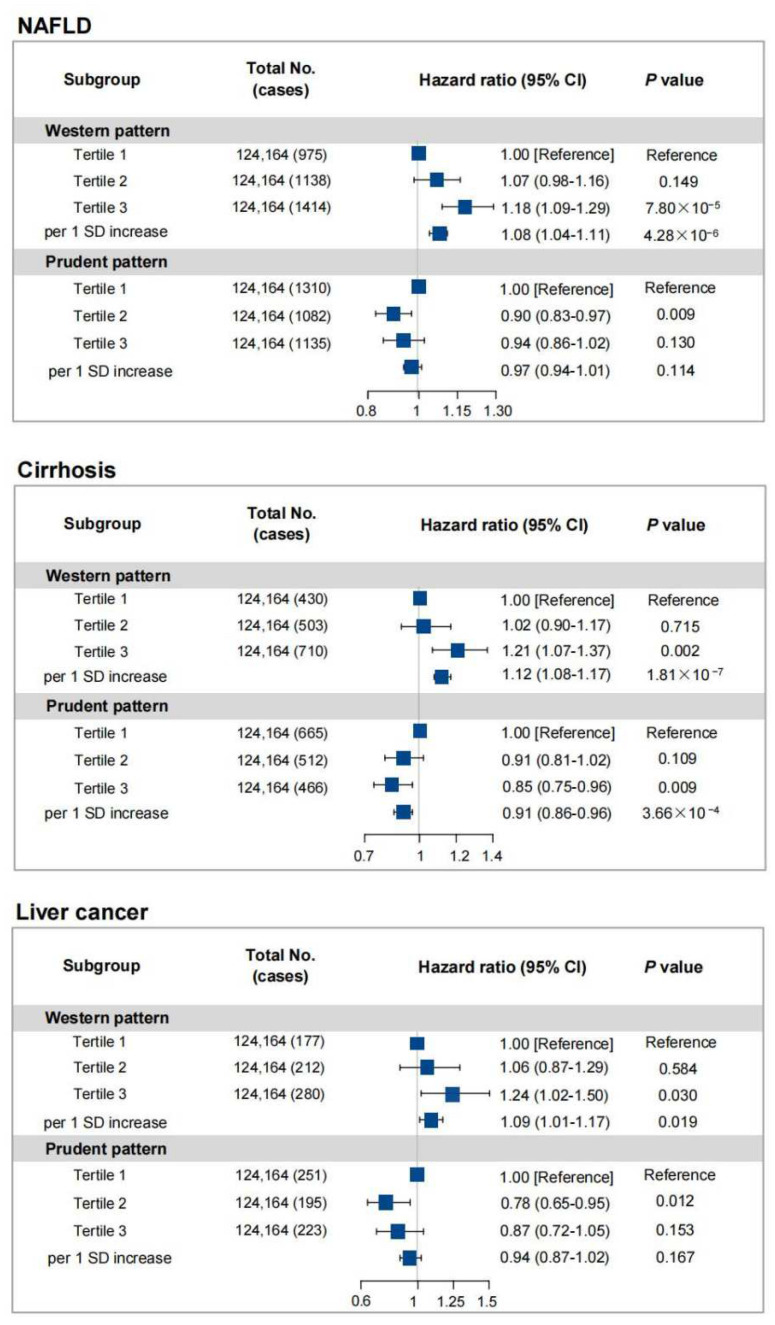
Hazard ratios for the risks of different liver diseases by dietary patterns (tertiles) in UK Biobank. HRs and 95% CIs were adjusted for age, sex, race, education level, Townsend deprivation index (quartiles), drinking status, smoking status, exercise, BMI, and diabetes. Definition of abbreviations: NAFLD, non-alcoholic fatty liver disease; HR, hazard ratio; 95%CI, 95% confidence interval; SD, standard deviation.

**Table 1 nutrients-14-05335-t001:** Baseline characteristics of participants in UK Biobank.

Characteristic	Total (*n* = 372,492)	NAFLD		Cirrhosis		Liver Cancer	
		Case (*n* = 3527)	Non-Case (*n* = 368,965)	Case (*n* = 1643)	Non-Case (*n* = 370,849)	Case (*n* = 669)	Non-Case (*n* = 371,823)
**Age (years), mean (sd)**	56.73 (8.10)	57.22 (7.88)	56.73 (8.10)	59.14 (7.44)	56.72 (8.10)	61.84 (6.18)	56.73 (8.10)
**Gender, %**							
Male	176,327 (47.34)	1751 (49.65)	174,576 (47.32)	1105 (67.26)	175,222 (47.25)	412 (61.58)	175,915 (47.31)
Female	196,165 (52.66)	1776 (50.35)	194,389 (52.68)	538 (32.74)	195,627 (52.75)	257 (38.42)	195,908 (52.69)
**White ethnicity, %**							
Yes	355,879 (95.54)	3342 (94.75)	352,537 (95.55)	1589 (96.71)	354,290 (95.53)	650 (97.16)	355,229 (95.54)
No	16613 (4.46)	185 (5.25)	16428 (4.45)	54 (3.29)	16559 (4.47)	19 (2.84)	16594 (4.46)
**Education level, %**							
College or University degree	133,579 (35.86)	874 (24.78)	132,705 (35.97)	411 (25.02)	133,168 (35.91)	182 (27.20)	133,397 (35.88)
Other levels	238,913 (64.14)	2653 (75.22)	236,260 (64.03)	1232 (74.98)	237,681 (64.09)	487 (72.80)	238,426 (64.12)
**Townsend deprivation index, %**							
Quartile 1	936,65 (25.15)	675 (19.14)	92,990 (25.21)	308 (18.75)	93,357 (25.17)	152 (22.72)	93,513 (25.15)
Quartile 2	925,55 (24.85)	722 (20.47)	91,833 (24.89)	320 (19.48)	92,235 (24.87)	164 (24.51)	92,391 (24.85)
Quartile 3	932,11 (25.02)	849 (24.07)	92,362 (25.03)	386 (23.49)	92,825 (25.03)	160 (23.92)	93,051 (25.03)
Quartile 4	930,61 (24.98)	1281 (36.32)	91,780 (24.87)	629 (38.28)	92,432 (24.92)	193 (28.85)	92,868 (24.98)
**Regular exercise, %**							
Yes	202,347 (54.32)	1623 (46.02)	200,724 (54.40)	751 (45.71)	201,596 (54.36)	336 (50.22)	202,011 (54.33)
No	170,145 (45.68)	1904 (53.98)	168,241 (45.60)	892 (54.29)	169,253 (45.64)	333 (49.78)	169,812 (45.67)
**Drinking status, %**							
Never	13,726 (3.68)	205 (5.81)	13,521 (3.66)	67 (4.08)	13,659 (3.68)	29 (4.33)	13,697 (3.68)
Former	11,802 (3.17)	225 (6.38)	11,577 (3.14)	123 (7.49)	11,679 (3.15)	23 (3.44)	11,779 (3.17)
Current	346,964 (93.15)	3097 (87.81)	343,867 (93.20)	1453 (88.44)	345,511 (93.17)	617 (92.23)	346,347 (93.15)
**Smoking status, %**							
Never	205,007 (55.04)	1581 (44.83)	203,426 (55.13)	626 (38.10)	204,381 (55.11)	278 (41.55)	204,729 (55.06)
Former	130,392 (35.01)	1424 (40.37)	128,968 (34.95)	702 (42.73)	129,690 (34.97)	295 (44.10)	130,097 (34.99)
Current	37,093 (9.96)	522 (14.80)	36,571 (9.91)	315 (19.17)	36,778 (9.92)	96 (14.35)	36,997 (9.95)
**BMI, %**							
Normal	126,020 (33.83)	357 (10.12)	125,663 (34.06)	317 (19.29)	125,703 (33.90)	162 (24.22)	125,858 (33.85)
Overweight	160,163 (43.00)	1286 (36.46)	158,877 (43.06)	591 (35.97)	159,572 (43.03)	264 (39.46)	159,899 (43.00)
Obesity	86,309 (23.17)	1884 (53.42)	84,425 (22.88)	735 (44.74)	85,574 (23.08)	243 (36.32)	86,066 (23.15)
**Diabetes, %**							
Yes	17,869 (4.80)	588 (16.67)	17,281 (4.68)	346 (21.06)	17,523 (4.73)	121 (18.09)	17,748 (4.77)
No	354,623 (95.20)	2939 (83.33)	351,684 (95.32)	1297 (78.94)	353,326 (95.27)	548 (81.91)	354,075 (95.23)

Definition of abbreviations: NAFLD, nonalcoholic fatty liver disease; BMI, body mass index; SD, standard deviation.

**Table 2 nutrients-14-05335-t002:** Factor loadings for dietary patterns in the UK Biobank.

Components	Prudent Pattern	Western Pattern
Salad raw vegetables	**0.64**	0.02
Cooked vegetables	**0.57**	0.12
Fresh fruit	**0.59**	−0.07
Dried fruit	**0.43**	−0.07
Oily fish	**0.52**	0.08
Non-oily fish	**0.39**	0.16
Processed meat	**−0.31**	**0.49**
Poultry	0.10	**0.44**
Beef	−0.09	**0.64**
Lamb/mutton	0.02	**0.66**
Pork	−0.05	**0.66**
Bread	−0.24	0.16
Cheese	−0.14	0.03
Tea	0.01	0.08
Cereal	0.18	−0.13
Variance explained [%]	13	12
Cumulative Var [%]	13	25

Using orthogonal rotation, correlation coefficients are nearly identical to the factor loading matrix. Bold denotes the absolute value of rotated factor loading ≥ |0.30|.

## Data Availability

The study data are available from UK Biobank (https://www.ukbiobank.ac.uk/), accessed on 28 May 2022. Restrictions apply to the availability of these data, which were used under license for the current study (Project ID: 85248). Data are available for bona fide researchers upon application to the UK Biobank.

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
