# Peer review of "Diet and Risk of Non-Alcoholic Fatty Liver Disease, Cirrhosis, and Liver Cancer: A Large Prospective Cohort Study in UK Biobank"

_nutrients, 2022, doi:10.3390/nu14245335_

Round 1

Reviewer 1 Report

The authors estimated the risk of diet on the development of liver disease. These results are not novel and some significant points need to be improved.

1. Subject characteristics 

B-viral and C-viral hepatitis are essential risk factors for HCC development. But the authors did not exclude viral hepatitis subjects in this study. 

2. Liver disease definition 

According to the supplemental table 3, the authors include alcoholic liver disease in the liver-related outcomes. NAFLD, Alcoholic liver disease, and viral hepatitis must be separated. 

Reviewer 2 Report

The presented paper is well dealing with a hot topic in liver disease how dietary patterns influence liver disease. The paper is well written and quite a bit of effort has been made to deal with the many cofunders of lifestyle patterns and diet. The data show for some componants of the western diet a negative effect for some components of the prudent diet a protective effect.

I think one important component is missing and that is coffee. Coffee has been associated with beneficial effects in alcoholic liver disease. It would be interessting to know how coffee influences the results if this is possible.  Another factor which is important is the intake of sugar in soft drinks and sweets. Especially in NAFLD this is ean important factor.

Second: It is not clear what cirrhosis mean, is it cirrhosis from all causes including autoimmune liver disease etc. or is the diagnosis related to NAFLD as the only cause.
